# Intra-Operative Electron Radiation Therapy: An Update of the Evidence Collected in 40 Years to Search for Models for Electron-FLASH Studies

**DOI:** 10.3390/cancers14153693

**Published:** 2022-07-29

**Authors:** Felipe A. Calvo, Javier Serrano, Mauricio Cambeiro, Javier Aristu, Jose Manuel Asencio, Isabel Rubio, Jose Miguel Delgado, Carlos Ferrer, Manuel Desco, Javier Pascau

**Affiliations:** 1Department of Oncology, Clínica Universidad de Navarra, 28027 Madrid, Spain; fserranoa@unav.es (J.S.); mcambeiro@unav.es (M.C.); jjaristu@unav.es (J.A.); 2Department of General Surgery, Hospital General Universitario Gregorio Marañón, Gregorio Marañón Complutense University, 28027 Madrid, Spain; josemanuel.asencio@salud.madrid.org; 3Gregorio Marañón Sanitary Research Institute (IiSGM), 28027 Madrid, Spain; 4Department of Surgery, Clínica Universidad de Navarra, 28027 Madrid, Spain; irubior@unav.es; 5Department of Medical Physics, Clínica Universidad de Navarra, 28027 Madrid, Spain; jdelgadorod@unav.es; 6Institute of Oncology, Hospital Provincial de Castellón, 12002 Castellon, Spain; carlos.ferrer@hospitalprovincial.es; 7Hospital General Universitario Gregorio Marañón, Department Bioengineering and Aerospace Engineering, Universidad Carlos III de Madrid, 28027 Madrid, Spain; desco@hggm.es; 8Centro Nacional Investigaciones Cardiovasculares Carlos III (CNIC), 28027 Madrid, Spain; 9Centro de Investigación en Red en Salud Mental (CIBERSAM), 28027 Madrid, Spain; 10Department of Bioengineering and Aerospace Engineering, Universidad Carlos III de Madrid, 28027 Madrid, Spain; jpascau@ing.uc3m.es

**Keywords:** IORT, IOeRT, electrons, FLASH, radiotherapy, intraoperative

## Abstract

**Simple Summary:**

Four decades ago, intraoperative electron radiation therapy (IOeRT) was developed to improve precision in local cancer treatment by combining real-time surgical exploration and resection with high-energy electron irradiation. The technology of ultra-high dose rate electron and other radiation beams known as FLASH irradiation sharply increases its interests, as data from preclinical experiments have proven a marked favorable effect on the therapeutic index: similar cancer control with a clearly improved tolerance of many normal tissues to high doses of irradiation. The knowledge and tools regarding technology, physics, biology, and preclinical results in heterogeneous cancers opens great opportunities towards the path of developing the first clinical applications of the emerging FLASH technology via clinical trials based on state-of-the-art medical practice with IOeRT.

**Abstract:**

Introduction: The clinical practice and outcome results of intraoperative electron radiation therapy (IOeRT) in cancer patients have been extensively reported over 4 decades. Electron beams can be delivered in the promising FLASH dose rate. Methods and Materials: Several cancer models were approached by two alternative radiobiological strategies to optimize local cancer control: boost versus exclusive IOeRT. Clinical outcomes are revisited via a bibliometric search performed for the elaboration of ESTRO/ACROP IORT guidelines. Results: In the period 1982 to 2020, a total of 19,148 patients were registered in 116 publications concerning soft tissue sarcomas (9% of patients), unresected and borderline-resected pancreatic cancer (22%), locally recurrent and locally advanced rectal cancer (22%), and breast cancer (45%). Clinical outcomes following IOeRT doses in the range of 10 to 25 Gy (with or without external beam fractionated radiation therapy) show a wide range of local control from 40 to 100% depending upon cancer site, histology, stage, and treatment intensity. Constraints for normal tissue tolerance are important to maintain tumor control combined with acceptable levels of side effects. Conclusions: IOeRT represents an evidence-based approach for several tumor types. A specific risk analysis for local recurrences supports the identification of cancer models that are candidates for FLASH studies.

## 1. Introduction

Cancer accounts for nearly 20% of deaths worldwide, representing nearly 10 million of the 55.4 million deaths worldwide [1]. The ratio between mortality and incidence varies globally a lot among countries based on tumor type distribution and performance of the healthcare system based on patient- and tumor related characteristics and cancer management innovation to tailor diagnostic and treatment into more precise medical treatment. Moreover, close multidisciplinary collaboration facilitates the search for an optimal blend of different treatments to obtain higher rates of tumor control while sparing organs and improving quality of life. IOeRT (Intraoperative electron Radiation Therapy) is a component of precise irradiation evidence based in multidisciplinary oncology. 

Preclinical developments made a breakthrough in 2014, when the first publication about very promising preclinical results obtained with ultra-high dose rate electron beams, called “FLASH-RT”, attracted a huge amount of attention worldwide [2] due to an unexpected reduction of toxicities in normal tissue compared to conventional radiotherapy, while still achieving local tumor control. Confirmed by different groups and on many preclinical models [3,4], the FLASH effect is defined as the combination of a relative absence of normal tissue toxicities compared to isodose of conventional dose rate RT combined with maintained anti-tumor efficacy. It has been observed after exposure of biological tissues to high doses in extremely short treatment times and with specific beam parameters including mean and instantaneous dose rates [5] mainly, using electrons [6], but also X-rays [7] and protons [8]. As transferring X-ray- and proton-FLASH into the clinics encounters numerous and important technological challenges, FLASH-RT using electrons is the logical first choice to being investigated for the transition from preclinical research to the first clinical applications. For this, both superficial tumors as well as deeper-seated tumors in the context of IOeRT are possible targets. The technical and practical ability to deliver electron FLASH-RT intraoperatively offers a new and fascinating challenge to further explore improvements of therapeutic interventions.

Integrated management of human cancer in the 21st century requires a complete and broad consensus within the oncology community on a coordinated interdisciplinary approach, as well as established guidelines [9]. The most advanced surgical, medical, and radiation oncology requires sustained MTB (Multidisciplinary Tumor Board) enrichment [10]. 

Breast cancer is a disease model with continuous innovation in local treatment and associated improvement [11,12]. Similar clinical and biological relationships have been demonstrated or expected for many cancer subtypes, based both on locoregional progression and systemic progression of the cancer. Presently, surgically guided RT is the best option for guided real-time irradiation. IOeRT allows delivering the required dose to post-resected high-risk target volumes. Displacement from the electron beam of dose-sensitive normal tissues uninvolved by cancer is major protective maneuver during surgical procedure. Since obtaining local control in cancer therapy is an essential requirement of achieving long-term survivorship with maintained functional normal tissues, real-time radio-surgical collaboration, a stunning new feature of precise radiation oncology technology exemplified by IOeRT, is highly beneficial to achieving long-term survivorship [13].

In an effort to improve patient safety in practice [14], technological innovation has led to the implementation of simulation and treatment planning systems [15] and in vivo dosimetry has been explored assessing real-time dose-delivering in clinical scenarios [16,17] or based in interactive Monte Carlo algorithms estimated in phantoms [18]. Failure mode and effect analysis (FMEA) tested in the IOeRT clinical process has the potential to reduce risk and improve quality [19].

Progress in surgery is adaptable to progress in IOeRT. This model of laparoscopic surgery is fully compatible with IOeRT (with locally advanced rectal cancer as a comparison) [20]. The feasibility of combining robotic surgery technology with IOeRT procedures with miniaturized-mobile electron linear accelerators has been tested and subsequently clinically successfully applied (Figure 1) [21]. 

Surgical navigation is feasible during open surgical procedures [22] (Figure 2a). Imaging advances will make it easier to guide radio-surgical manoeuvres during intra-planning, and to register and document technical parameters in real time using ultrasound (Figure 2b).

In academic expert institutions, a special interest is generated for the development of normalized clinical practice based on prospective data recording and e-learning resources [23].

## 2. Material and Methods

### IOeRT: 40 Years of Clinical Results: Evidence-Based Data from IOeRT in 6 ESTRO/ACROP Guidelines 2020

The performance and quality of intraoperative radiation therapy (IORT) publications identified in medical databases during a recent period in terms of bibliographic metrics has been reported [24]. An updated extensive bibliometric search revealed a total of 19,148 patients registered in 116 publications evaluated analyzing publications up to 2020 (period 1982–2020) (Figure 3). 

An expert-based task force under the umbrella of the European Society for Radiotherapy and Oncology (ESTRO) condensed the available data in combination with expertise/expert opinion into a total of 6 guidelines concerning soft tissue sarcomas, unresected and borderline-resected pancreatic cancer, locally recurrent and locally advanced rectal cancer, and breast cancer. The ESTRO/ACROP (Advisory Committee for Radiation Oncology Practice) recommendations of all guidelines included the background and requirements for every aspect of clinical IOeRT practice, including patient selection, diagnostic and therapeutic procedures, quality assurance, and reporting [25,26,27,28,29,30]. 

The breast cancer IOeRT model is the most numerous in terms of registered patients (4229 in 12 studies on IOeRT as a partial breast irradiation technique with a single fraction and 4414 patients reported in 9 publications using IOeRT as an anticipated boost, both after breast conserving surgery). The chronology of breast cancer data is peculiar: it is the last actor in published recording (no relevant publications are available before the year 2000), but in the last two decades, the growth of IOeRT-related publications has been exponential [25].

On the other side, pancreatic cancer is the clinical model that is better documented in the early publications (1980s and 1990s) with a sustained bibliometric representation along 40 years (total of 2307 patients described in 36 publications with unresected localized disease and 2087 post-resected patients in 33 papers) [26,27]. 

The case of rectal cancer is particularly relevant due to the fact that it contains a significant proportion of patients treated with IOeRT for the rescue of oligo-recurrent disease (total of 1730, 10 publications). A remarkable finding from these data is the existence of a quite large and constant group of patients that remain alive and controlled, over long time periods, something that was not reported previously with other surgical or RT techniques [28]. The integration of an IOeRT component of treatment in the combined management of locally advanced primary rectal cancer (2590 patients, 18 studies) is a successful extension of the adaptation of IOeRT boost to further increase local control in a cancer site that has witnessed the introduction of multiple multimodality treatment approaches through the course of the four decades that we analyzed [29].

The last guideline to be mentioned refers to the use of IOeRT in the management of soft tissue sarcomas which has two well-defined populations: extremity (922 patients in 14 reports) and retroperitoneal (871, 24). Sarcoma publications appeared only in the 21st century and the increment in patients reported is superior in the extremity involvement model using IOeRT as an anticipated boost [30].

## 3. Results

Clinical outcomes following IOeRT doses in the range of 10 to 25 Gy (with or without external beam fractionated radiotherapy) show a wide range of local control ranging from 40 to 100%, depending upon cancer site, histology, stage, and treatment intensity (Table 1). This information is essential for identifying opportunities for de-escalation as well as for intensification and identify candidates for FLASH IOeRT clinical trials.

## 4. Discussion

### 4.1. IOeRT Radiobiology: The Conundrum of Boosting versus Single High-Dose Fraction

Patient selection for IOeRT is done based on a number of patient-, tumor-, and treatment-related factors. In this, preoperative imaging is of tremendous importance. However, the relevant factors to precisely guide dosimetric decisions in IOeRT are highly dependent on conditions of the biological and clinical target volume implemented by the delivery through a wide set of collimators that are available in many sizes and bevel angles.

In IOeRT, the high precision in dose delivery allows for the exploration of single fractions of high doses of radiation, facilitated by normal tissues being protected by mechanical displacement and/or intraoperative shielding. The tolerance of normal tissues has been meticulously explored in large animal models using single escalated doses in the range of 10 to 40 Gy [31], or in a range of escalated boost levels from 10 to 30 Gy combined with a component of fractionated external beam RT (generally 50 Gy in 25 fractions). Based on this, combined with human data, general recommendations include a dose of 20 to 25 Gy as a single dose or 10 to 15 Gy as a boost dose after 50 Gy fractionated external beam RT [32]. In the intraoperative scenario, common features are low cancer cell burden, high growth factor content, and the presence of repair and inflammatory responses. Despite this, long-term clinical therapeutic indexes are only available based on historical data without further modeling. The equivalent dose in 2-Gy fractions (EQD2) model has been shown to predict cancer control in patients who have been rescued from recurrent sarcoma, and to show a survival benefit for EQD2 doses above 62 Gy [33]. New hypotheses to individualize clinical practice using information from retrospective large and mature datasets require revisiting basic parameters (such as dose, volume, and fractionation) in the context of bio-molecular profiles and signatures.

The unique experience using exclusively 21 Gy IOeRT in localized breast cancer has provided pioneering information of clinical correlations between biological and clinical risk factors and local cancer control [34]. ASTRO and the GEC-ESTRO recommendations have established the eligibility criteria for partial breast irradiation (APBI). In this large cohort of patients, matched according to the ASTRO and the GEC-ESTRO low-risk classification, a 5-year 1.5% and 1.9% recurrence rate was observed [35,36]. In the randomized ELIOT trial, a group of women at a very low recurrence risk was defined based on four concomitant characteristics: tumor size < 1 cm, histological grade 1, luminal A molecular subtype and proliferative index (Ki-67) < 14% [37]. Recently, the ESTRO/ACROP consensus recommendations on patient selection and dose and fractionation for external beam radiotherapy in early breast cancer revised and finetuned the patient selection criteria for partial breast RT [38]. 

### 4.2. IOeRT: Roll-Out of the Clinical Applications

Today, IOeRT clinical practice applications are widely documented in the scientific literature and specific recommendations are available for data recording and innovation needed to adapt changing practices in the combination of cancer surgery and IOeRT [23]. This large and mature body of evidence demonstrates the feasibility, tolerance, and benefits in terms of cancer control obtained by using IOeRT as a component of treatment in patients requiring surgery and RT. It also constitutes a solid evidence-based strategy for future developments. Surgical and radiation technology innovations have methodological constraints. Indeed, the inherent difficulties related to proper health technology assessment as seen in several diagnostic and therapeutic developments is well-known in radiation oncology [39]. Regardless, future development of IOeRT will need to be built on the principles of precision medicine and personalized oncology, in the framework of proper and solid clinical research. As part of the integrated radiation and surgical treatment scenario, technological developments are already being implemented, or are in development, to further improve precision in dose delivery, dose measurement, dose registration, and dose integration with other treatment factors [40]. Individualized risk-adaptive treatment recommendations can be structured from clinical heterogeneity by incorpating IOeRT [41]. Therefore, NCCN guidelines incorporated IOeRT as a part of the treatment for several cancer sites (http://www.nccn.org, accessed on 31 January 2020). 

As today only a small proportion of the RT departments has the required infrastructure, staffing, and knowledge for delivering high-quality IOeRT, it is the role of the existing IOeRT community to build networks and to assist in the roll-out of IOeRT to more departments to obtain full coverage of the obvious healthcare demands for IOeRT.

### 4.3. IOeRT: Research and Clinical Perspectives with FLASH-IOeRT

The FLASH effect is expected to be a major game changer in radiation oncology, thanks to the promises of sparing normal tissue late toxicities without a decreased tumor control effect [2]. As IOeRT is most likely the first application to be able to clinically implement FLASH-RT. The setup of IOeRT is less influenced by those current technical uncertainties around the application of FLASH-RT in the clinical, as the target conformality dependence on achieving the FLASH effect, the use of multi angle beam settings, and a treatment planning system adapted to UHDR able to simulate the effect of beam settings per small volumes/voxels. 

Additionally, IOeRT is potentially very widely available; the time is there to work out topics for preclinical research that facilitate early transition to possible clinical applications. Those should be based on tolerance of normal tissues, with an objective to maintain tumor control combined with fewer side effects for indications with high local control rates (e.g., breast cancer), while for indications representing resistant or unresectable disease (e.g., oligo-recurrent status, sarcomas, and pancreas cancer), the objective should rather be to further improve cancer control through radiation intensification while maintaining acceptable levels of side effects.

The single dose electron delivery is a known experimental model, multiple preclinical studies have showed that large single doses of electron FLASH-RT induce less early and late normal tissue toxicities than similar doses delivered at conventional dose rates. This was particularly observed for encephalic, thoracic, abdominal, and skin exposures.

It is important to remark the volume effect, most preclinical results on FLASH-RT have been obtained on small mammals, mostly mice, thus using small irradiation fields (<2–3 cm diameter).

Apart from the established indications for IOeRT, including breast, rectal, pancreas and sarcoma, tumor sites such as skin, brain, and pediatrics are also conceptually attractive [7,42]. For each of them, a specific risk analysis combining local tumor control and side effects should be made to support the selection of the most appropriate cancer models for future FLASH research. A proposal for dose-escalation clinical studies is presented in Table 2.

## 5. Conclusions

Intraoperative radiation therapy using high energy electron as radiation beam has a mature and in-depth reported dataset from over the last 40 years. Results in terms of cancer control and normal tissue tolerance are subject to further improvement. Several evidence- and consensus-based guidelines for clinical IOeRT applications are available. The emerging technology of FLASH irradiation promises to exert a marked favorable effect on the therapeutic index by maintaining cancer control while improving tolerance of many normal tissues to high doses of irradiation. As the first clinical applications of FLASH irradiation are by far most feasible and straightforward via IOeRT, a significant rise in the interest for IOeRT emerges. Clinical introduction requires setting up a large number of clinical trials, combining state-of-the-art medical practices of IOeRT with the emerging FLASH-technology.

## Figures and Tables

**Figure 1 cancers-14-03693-f001:**
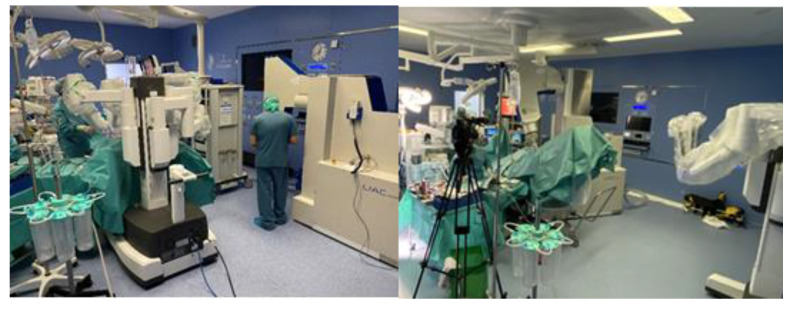
A modern operating room with a miniaturized-mobile electron linear accelerator and a robotic Da Vinci system used together to treat a prostate cancer patient with IOeRT (post-resection of oligonodal relapse) Liac HWL, Sordina IORT Technologies.

**Figure 2 cancers-14-03693-f002:**
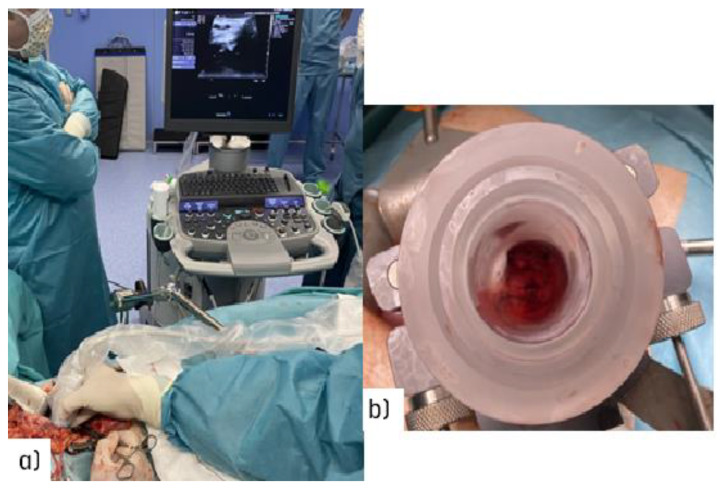
Illustration of the target definition imaging procedure during an open abdominal procedure: (**a**) real-time ultrasound assessment of the coeliac trunk with an unresectable nodal recurrence; (**b**) view through the electron beam applicator during the procedure in a patient with recurrent gastric cancer (uninvolved normal sensitive tissues in the upper abdomen are displaced out of the IOeRT target volume).

**Figure 3 cancers-14-03693-f003:**
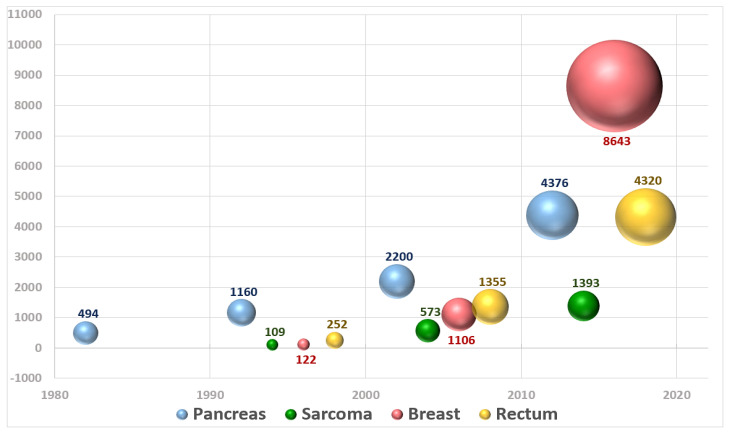
Graphical representation of the number of patients included in publications about IOeRT per year over the period 1982–2020.

**Table 1 cancers-14-03693-t001:** Local control and survival rates in 19,148 patients analyzed and reported in the ESTRO/ACROP guidelines by cancer site and IOeRT dose (period 1982–2020). Abbreviations: # (number); pts (patients); REF (references); IOERT: intraoperative electron radiation therapy; Gy (Gray); OS (overall survival); y (years).

Cancer Site/Type/Status	# pts (%)	# REF (%)	IOERT DOSE Gy	Local Control	OS 5y
Pancreas Unresected	2307 (12%)	36 (31%)	15–25	41–71%	0–6%
Pancreas Resected	2087 (10%)	33 (28%)	10–20	73–94%	20–35%
Rectal Locally Advanced	2590 (13%)	18 (15%)	10–15	75–100%	64–84%
Rectal Oligo-Recurrent	1730 (9%)	10 (8%)	12.5–20	44–72%	25–52%
Soft Tissue Sarcoma Extremity	922 (5%)	14 (12%)	10–20	58–100%	69–82%
Soft Tissue Sarcoma Retroperitoneal	871 (4%)	24 (20%)	10–20	40–90%	38–74%
Breast Cancer (Partial Breast RT)	4229 (22%)	12 (10%)	21–23	91–100%	94–100%
Breast cancer (BOOST)	4414 (23%)	9 (7%)	10–15	89–100%	75–97%

**Table 2 cancers-14-03693-t002:** FLASH dose-escalation proposal guided by ESTRO guidelines dose recommendations in several cancer models. Methodology based on phase l–II oriented studies with increments of 5 Gy is considered with electron FLASH beams over the conventional higher dose recommended by cancer type and post-surgical disease and margin status.

IOeRT ESTRO Cancer Models Guidelines (Ref.)	ESTRO Dose IOeRT Recommendation (Gy)	Normal Tissues at Risk within IOeRT Target	FLASH Dose Escalation Proposal (Gy)
Unresected pancreas [27]	15–20	pancreatic tumor and parenchyma bile duct duodenum vascular structures vertebrae	25–30–35
Post-resected pancreas [26]	R0 10–12.5 R1 12.5–15 R2 15–20	vascular structures vascular sutures * retroperitoneal tissue	R0 17.5–20–25 R1 20–25–30 R2 25–30
Extremity sarcomas [30]	R0 10 R1 15 ** R2 20 **	peripheral nerves muscle vascular structures bone	R0 15–20–25 R1 20–25–30 R2 25–30
Retroperitoneal sarcomas [30]	Close margins 10–12.5 Involved margins 12–15 Gross residual 15–20	peripheral nerves muscle vascular structures bone ureter * vascular suture *	Close margins 15–20–25 Involved margins 20–25–30 Gross residual 25–30
Primary advanced rectal cancer [29]	R0 10–12.5 R1 12.5–15 R2 15–20	bone peripheral nerves vascular structures soft tissue	R0 17.5–23–28 R1 20–25–30 R2 25–30
Locally recurrent rectal cancer [28]	R0 12.5–15 R1 15–20 R2 15–20	bone peripheral nerves vascular structures soft tissues vascular sutures * ureter previous irradiated tissues *	R0 17.5–23–28 R1 20–25–30 R2 20–25–30 * 20–25–30
Breast cancer partial irradiation [25]	21	breast parenchyma reconstructed	26–30
Breast cancer boost [25]	9–12	breast parenchyma reconstructed chest wall components: - Intercostal muscle - Intercostal nerve - Intercostal vessels pleura *	17–22–30

* Potential. ** Field-in-field technique.

## Data Availability

The datasets used and/or analyzed during the current study are available from the corresponding author on reasonable request.

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
