# Peer review of "Intra-Operative Electron Radiation Therapy: An Update of the Evidence Collected in 40 Years to Search for Models for Electron-FLASH Studies"

_cancers, 2022, doi:10.3390/cancers14153693_

Round 1

Reviewer 1 Report

This is an excellent review of the present evidence for the efficacy of IORT electron treatments. As IORT electron treatment tend to consist of a single delivery of a high (>20 Gy) amount of radiation, they seem particularly well-suited to serve as a launching point for the original fractionations to be used in FLASH-RT, which up to now has mostly been observed in very large single fractions of radiation.

The only caveat I would bring is that the paper should do more to explain the link between the proposed review and the lessons that can be learned in implementing FLASH-RT to the clinic. The authors clearly make the link in their title, and in their introduction, but the concepts are not sufficiently explored in the discussion. Greater comparison between specific pre-clinical or veterinary studies and past IORT regimen may be in order, as well as more specific lessons the authors suggest we may learn from IORT when implementing FLASH in the clinic.

Author Response

  • Comment "link to implement FLASH-RT": table 2 (line 307) has been introduce 
  • Comment "preclinical veterinarian studies": table 2 (line 307) covers in part this information as a proposal for dose implementation based in clinical tolerance (and veterinarian studies)

Reviewer 2 Report

In this study, the authors reviewed publications in the past 40 years that reported clinical outcomes using intra-operative electron-beam radiation therapy to different cancer sites. Based on the title of this manuscript, the apparent aim of this study is to "search for models for electron-FLASH studies". However, this manuscript failed to show that the results from this study achieved this aim. In fact, the Results section of this manuscript only contains a table that summarizes radiation therapy treatment and clinical outcomes data for a list of studies from literature search. Since all the publications that were reviewed in this study did not use the FLASH technique, it is a leap of faith to arrive at any conclusion related to FLASH therapy based on a review of those publications. The current manuscript does not provide any new or useful information to the radiation oncology community and does not make useful predictions on using the FLASH therapy.

Specific comments:

1. Line 200: "local recurrences" probably should be "local control" based on Table 1 data.

2. Line 41: Again, should "local recurrences" be "local control" based on Table 1?

3. Since the authors discussed using IOeRT as either boost or single fraction treatment, I recommend that the authors provide a comparison of clinical outcomes with each of the two IOeRT treatment options in the Results section. 

4. Please define the abbreviations used in Table 1.

Author Response

  • comments "failed to show results": No clinical results are available with FLASH electron beams used intraoperatively.
  • "no new usefull information": this is the largest set of patient data treated with intraoperative electron irradiation organized with chronologic metrics (40 years).
  • "no usefull predictors FLASH": FLASH animal models are referenced but not extrapolated into clinical considerations on purpose (animal sizes, tissue characteristics and radiation doses are not an appropiate source for prediction)
  • line 200: has been corrected as suggested
  • line 41: has been corrected as suggested
  • Table 1 abreviations: have been added
  • "boost vs single fraction": outcomes are described in table 1 separately included in the Results section

Reviewer 3 Report

This article reviewed publications regarding IOeRT and discussed potential of IOeRT using FLASH-RT. This article showed that IOeRT can be coupled with minimally invasive surgery such as robotic surgery. As discussed this article, IOeRT can be applied for breast cancer, STS, pancreatic cancer, and rectal cancer.

I think this article may have readers' interest of Cancers.

I have just following minor concerns.

1. Provide full words for IOeRT and MTB in main test.

2. Subtitle on 98th line on 3rd page is not necessary.

3. Check local recurrence rates on 200th line on 6 page. These values do not matched with Table 1.

Author Response

  • IOERT: intraoperative electron radiation therapy (is describe in line 21, 31 and 59)
  • MTB: Multidisciplinary Tumor Board (introduced in line 83 as suggested)
  • line 98: subtytle removed as suggested
  • line 202: data corrected as suggested

Reviewer 4 Report

This work represents a good synthesis of the state of the art in the use of IOeRT over 40 years:

it correctly describes the clinical applications of the technique and the results obtained in local control and survival rate in the various pathologies treated.

The article also underlines the benefit obtained in clinical application of IOeRT, thanks to the creation of International Guidelines.

Furthermore, the work inserts in the argumentation  the recent variable of personalized medicine, necessary to pursue an improvement in care: in this scenario it introduces the possible use of FLASH-RT as a further therapeutic opportunity.

In the work, the authors underline the importance of having specific technologies and accurate skills, hoping for the creation of a network of activities, suitable to play a leading role in the knowledge and future applications of the technique; they also   call for the identification of specific radiobiological models and emphasize the importance of a particular focus on cancer biology, to design optimal FLASH-RT therapeutic protocols.

On the other hand, the work does not describe, but only recalls, the characteristics of the FLASH-RT through the indication of precise and pertinent bibliographic references previously produced:

in this regard it could be useful, for a full use of the article, to insert a brief synthesis of the technological, dosimetric and radiobiological FLASH-RT characteristics and to report the main preclinical experiences produced with it, even if only in a summary table.

Author Response

  • Comment "does not describe but recalls FLASH": available data is based upon animal experiments and requires a full redesign for clinical extrapolation
  • "Summary table for FLASH radiobiological results": traslational results from animal experimental FLASH data is concerning. A new table (table 2) has been added with a conservative proposal for dose-escalation studies based on clinical judgement

Reviewer 5 Report

The manuscript seems to be quite similar to a previous publication of the same author, named: ‘Intraoperative irradiation: precision medicine for quality cancer control promotion’ (DOI:10.1186/s13014-017-0764-5), with some parts being identical.  The authors should revise the manuscript, otherwise it, would be deemed plagiarized. 

·       The flow of the manuscript is not easy for reading, without linking of the information within the manuscript. In addition, there are parts where some phrases were repeated in the same sentence (see lines: 64-69, 91-95).

·       The authors mentioned in the manuscript (line: 23-24) ‘The technology of ultra-high dose rate electron beams known as FLASH irradiation’. FLASH radiotherapy is not implemented only with electrons. Please re-phrase. 

·       It is not mentioned if the figures (1&2) are from a personal archive or from any other department. This should be clarified. Otherwise, the authors should provide permission.

·       The authors have not made the analysis according to PRISMA guidelines.

·       The connection between IOeRT and FLASH is not clear in the manuscript. The authors should provide more details about the dose rate of IOeRT for further categorization in the analysis, since the publications reviewed include patients treated with IOeRT in a period of 40 years. The technological advancements in those years could have an impact in the clinical outcomes even if the treatment site was the same.

Author Response

  • Plagiarism. OED "... taking someone else´s work, idea,etc, and passing off as one´s own". The same author can not be involved in plagiarism (a level of intelectual property remains after communication of scientific ideas). 
  • lines 91-95 have been corrected
  • Line 23-24 have been re-phrase
  • Figures 1 and 2: are from a personal archive (fisrt manuscript author)
  • PRISMA guidelines: this is a chronologic literature review, (not systematic, no meta-analysis)
  • "dose rate of IOERT": in 40 years the technology has change. Along this time, linnear accelerators used in IOERT procedures were comercial equipments with relatively standard dose-rates. This information is not generaly available in the clinical publication consulted.

Round 2

Reviewer 2 Report

This revision answered the questions raised in the initial review. It is acceptable for publication.

Author Response

Thank you for your revision

Reviewer 5 Report

·       Regarding the remark about plagiarism.

    We thank the authors about the definition of plagiarism but OED provides the definition of autoplagiarism too.

 The authors have changed some parts of the manuscript by deleting them. However, there are still paragraphs that are almost identical to their previous paper  (DOI:10.1186/s13014-017-0764-5)

See lines:

103-111: same with ‘Research and innovation: precision for quality and safety’ section, 1st paragraph

139-144: same with ‘Research and innovation: precision for quality and safety’ section, 3rd  paragraph

223-228: same with ‘Radiobiology: intensification vs sub-intensification’ section, 1st paragraph

·       Lines 64-69 have not been rephrased, as suggested. It is not clear what the authors meant.

Did the authors mean that ‘the FLASH effect, reduces toxicities in normal tissue compared to conventional radiotherapy, while still achieving local tumor control’?

    The authors’ comment about the connection between IOeRT and FLASH is not adequate. The contribution of the FLASH technique and its clinical impact is not highlighted in the manuscript.

      Please give more information about FLASH technique and its possible clinical impact

Author Response

  • I appreciate the reviewer comment regarding "autoplagiarism". I did not know it existed. Thank you for this interesting personal lesson.
  • lines 103-111 Paragraph has been deleted 

    IOeRT: 40 years of research, innovation, and education efforts

    Biomedical engineering is capital for technological innovation, while research is the engine for increasing knowledge and education is essential for enabling high-quality medical practice in a safe environment (14). Departments that are leading in IOeRT applications have set up active programs to implement adapted simulation and treatment planning systems (15), participated in developing intraoperative imaging to further optimize treatment planning, and apply real-time in-vivo control of dose delivery in the clinical target volume. As a result, dosimetry instruments (16-18) have been successfully incorporated into multi-professional coordinated activities such as IOeRT, combine with risk models to analyze and estimate potential errors and promote patient safety.

  • and concepts have been reorganized: 

    "In an effort to improve patient safety in practice (14), technological innovation has led to the implementation of simulation and treatment planning systems (15) and in vivo dosimetry has been explored assessing real-time dose-delivering in clinical scenarios (16, 17) or based in interactive Monte Carlo algorithms estimated in phantoms (18). Failure mode and effect analysis (FMEA) tested in the IOeRT clinical process has the potential to reduce risk and improve quality (19)."

  • line 139-144 Several sentences have been deleted Publication based analysis showed 972 papers on IOeRT topics, of which 41% in surgical journals (period 1997–2013). The most frequently published topic was cancer clinical outcome (68%). The metrics of published evidence showed progressive significant  improvement on  the median impact factor over time 

New sentence has been added 

The performance and quality of intraoperative radiation therapy (IORT) publications identified in medical databases during a recent period in terms of bibliographic metrics has been reported (24).

  • line 223-228 several sentences have been deleted 

    Continuous teaching, education and training are key elements for implementing and maintaining high quality clinical practices, with further improvement in existing programmes to optimise the learning curve of IOeRT techniques in newly adopting institutions. New e-learning initiatives for sharing knowledge in clinical case-based problem-solving using educational platforms and dedicated websites are being implemented (23).

  • New sentence has been added 

    In academic expert institutions an special interest is generated for the development to normalized clinical practice based on prospective data recording and e-learning resources (23)

  • Line 223-228 Several sentences have been deleted . The tolerance of normal tissues has been meticulously explored in large animal models using single escalated doses in the range of 10 to 40 Gy (31), or in a range of escalated boost levels from 10 to 30 Gy combined with a component of fractionated external beam RT (generally 50 Gy in 25 fractions). Based on this, combined with human data, general recommendations include a dose of 20 to 25 Gy as a single dose or 10 to 15 Gy as a boost dose after 50 Gy fractionated external 
  • New sentence has been added 

    In academic expert institutions an special interest is generated for the development to normalized clinical practice based on prospective data recording and e-learning resources (23)

  • line 64-69 sentence delated absence of radiation-induced normal tissue toxicity after FLASH-RT compared to an isodose of conventional dose rate RT and coupled to a maintained anti-tumour.
  • sentence added as sugested reduction of toxicities in normal tissue compared to conventional radiotherapy, while still achieving local tumor control
  • connection of IOeRT and FLASH: the connection is based in the opportunity of further increase local cancer control  (by significant FLASH dose-delivering-escalation without significant normal tissue tolerance compromise). This is explain in the last paragraph (IOeRT: research and clinical perspectives with FLASH-IOeRT) to the level of available data possible inference and specific suggestions are made in table 2.

Round 3

Reviewer 5 Report

The revised  version of the manuscript has been
sufficiently improved. 

OK for publication